# Characteristics of Plantar Pressure with Foot Postures and Lower Limb Pain Profiles in Taiwanese College Elite Rugby League Athletes

**DOI:** 10.3390/ijerph19031158

**Published:** 2022-01-20

**Authors:** Tong-Hsien Chow, Yih-Shyuan Chen, Chin-Chia Hsu, Chin-Hsien Hsu

**Affiliations:** 1Department of Leisure Sport and Health Management, St. John’s University, New Taipei 25135, Taiwan; thchow1122@mail.sju.edu.tw (T.-H.C.); hill@mail.sju.edu.tw (C.-C.H.); 2Department of Education, National Pingtung University, Pingtung 900391, Taiwan; katy@mail.nptu.edu.tw; 3Department of Leisure Industry Management, National Chin-Yi University of Technology, Taichung 41170, Taiwan

**Keywords:** elite rugby league athletes, arch index (AI), plantar pressure distributions (PPDs), low-arched supinated feet, cuboid syndrome

## Abstract

Background: This study aimed to explore the differences in the distributions of plantar pressure in static and dynamic states and assess the possible pain profiles in the lower limb between elite rugby league athletes and recreational rugby players. Methods: A cross-sectional study of 51 college elite rugby athletes and 57 recreational rugby players was undertaken. The arch index (AI) and plantar pressure distributions (PPDs) with footprint characteristics were evaluated via the JC Mat. Rearfoot alignment was examined to evaluate the static foot posture. The elite group’s lower-limb pain profiles were examined for evaluating the common musculoskeletal pain areas. Results: The recreational group’s AI values fell into the normal range, whereas the elite group’s arch type fell into the category of the low arch. Results from the elite group were: (1) the PPDs mainly exerted on the entire forefoot and lateral midfoot regions in static standing, and transferred to the forefoot region during the midstance phase of walking; (2) the static rearfoot alignment matched the varus posture pattern; (3) the footprint characteristics illustrated the features of low-arched, supinated, dropped metatarsal heads and dropped cuboid feet; and (4) the phalanx and metatarsophalangeal joints, and the abductor hallucis and abductor digiti minimi of the plantar plate were common musculoskeletal pain areas. Conclusions: Characteristics of higher plantar loads beneath forefoot and midfoot associated with low-arched supinated feet in bipedal static stance could be the traceable features for the foot diagram of elite rugby league athletes. The limb pain profiles of the elite rugby league athletes within this study echoed the literature on rugby injuries, and reflected the features of metatarsophalangeal joint pains and dropped cuboids. The relationships among the low-arched supinated feet, metatarsophalangeal joint pains and cuboid syndrome are worth further studies.

## 1. Introduction

Overload of plantar pressure may lead to soft tissue degeneration, such as fat pads of the foot during exercise [1]. This may increase the risk of foot disease development in adults and children [2]. There is a potential link among the foot structure, lower-limb function, balance ability and obesity, due in particular to functional overload [3]. Biomechanical or anatomical changes and abnormalities of the skeletal joints represent risk factors that may be beneficial to accidents in many sport disciplines [4]. Specifically, the fifth metatarsal fractures are podiatric problematic for professional soccer athletes [5]. Muscle fatigue in footballers is believed to be at the core of developing metatarsal stress fractures because of the increased load on the forefoot of runners in a state of fatigue [6]. Proximal fifth metatarsal fractures, commonly referred to a ‘Jones Fracture’, is a common podiatric disease in basketball [7] and soccer [8] players, and usually occurs when the proximal fifth metatarsal suffers from the forces caused by the specific repetitious jumping and cutting [8,9].

In addition, cuboid syndrome is most likely to happen in people who participate in particular sports such as running, tennis, basketball, and ballet [10,11]. These sports exert a lot of force through the foot and rapid movement where players are quickly changing direction [11]. This, in turn, increases the pressure on the joints and attachments of the cuboid bone. Repeated pressure on the cuboid bone may cause the support around the cuboid bone to loosen, resulting in displacement and dislocation of the cuboid bone. [11]. Cuboid syndrome usually results from a flat foot when the foot arch is lowered. Abnormal foot arches may exert disproportionate stress to the cuboid bone and increase pressure through the muscles of the lower extremities [11]. Rugby is unavoidably one of the disciplines at risk of such problems [4]. 

Rugby is a sport of high-intensity contact and frequent strong collisions. Most rugby injuries were located in the lower extremities [12]. Carl et al. pointed out the fact that greater vertical forces, impact loading rates and peak plantar pressure occur when running in football boots compared to running or training shoes [13]. These high peak pressures are focused on the forefoot (i.e., hallux, medial, central and lateral forefoot) and steadily reduced at the heel during acceleration [14,15]. However, little rugby-specific research has been conducted for investigating fatigue-induced alterations in plantar foot-loading [6]. 

Plantar pressure detection is one of the effective methods for assessing plantar loads, seriousness of podiatry, abnormal gaits, rehabilitation condition [16] and ambulatory activities [17]. It could show the function of the feet and ankles in the gait cycle, because the feet and ankles provide the necessary support and flexibility for weight bearing and weight transfer during activities [18]. The parameters can be used to reveal the relationship between the multisegment, fine structure of the foot and the detailed function of the foot, [19] which is helpful for detecting foot pathology and warning, preventing, treating, rehabilitating and recurrence of foot deformities [20,21]. In addition, static plantar pressure measurement is considered to be helpful to solve many problems related to the relationship between plantar loads and lower-limb posture, thus, measurement techniques of plantar pressure are beneficial to understanding the biomechanics of the human foot [22]. 

Plantar pressure is mainly examined during walking and running in adults in current studies [23,24,25]. The parameters of plantar pressure and body posture in static state in elite athletes of different disciplines were discussed in previous studies [4,15]. Nevertheless, few studies have been undertaken for exploring the causal relationship between the etiology of the podiatric pain caused by the specific plantar pressure distribution and the special plantar pressure pattern induced by the rugby discipline. On this basis, the possible link between plantar pressure profiles and lower limb pain profiles related to rugby discipline is worth further exploration. This study aimed at establishing the overall reliability of the relationships among the arch index (AI), the plantar pressure distributions (PPDs) and the rearfoot postural alignment of college elite rugby athletes during static standing and walking. Given that the potential pain profiles in the athletes’ performance model are commonly related to sport per se, the elite rugby athletes’ pain areas and self-reported health status were assessed with a particular focus on the examination of the possible link between pressure profiles and lower-limb pain profiles, which were connected directly to rugby drills.

## 2. Materials and Methods

### 2.1. Participants

This cross-sectional study examined the relationships among the AI, the PPDs, the rearfoot postural alignment and the potential pain profiles of the lower extremity. Research participants comprised 108 male college and university students in Taiwan, who were categorized into two groups: 51 elite rugby league athletes (the elite group) and 57 healthy, age-matched recreational rugby players (the recreational group). The study was conducted during the noncompetition period, the inclusion criteria in the elite group were qualified first-division rugby athletes who had more than five successive years of rugby training experience in the National Rugby Championship, Taiwan Rugby Championship 7 s and Taiwan Chung-Cheng Cup-Senior High School Group. The elite rugby league athletes were between the ages of 19 and 22, and were registered in the Chinese Taipei Rugby Football Union (CTRFU). They were selected from 75 male elite rugby athletes who were recruited from St. John’s University, Chang Jung Christian University, Chinese Culture University, National Taiwan Normal University, National Taiwan Ocean University and Taipei Medical University in Taiwan. There was a 32% dropout rate in the process of recruiting the eligible elite rugby athletes for the following reasons: (1) absence rate; (2) a physician’s certificate of the past fracture or surgery; and (3) having taken or been taking any anti-inflammatory analgesics in the past one month. Rugby workout schedules (Monday to Friday), including basic movements and aerobic training, were set from 8 AM to 10 AM. Weight training and tactical training were set from 3 PM to 5 PM, and 1 to 2 hours of high-intensity interval sprint training was set for 3 to 5 days a week.

The eligible 57 recreational rugby players (the recreational group) within this study were selected from 78 rugby players who had more than 4 years of recreational rugby experience and played rugby at least 2 days per week at a rugby pitch or sports field within 6 months before this study was initiated. There was approximately a 27% dropout rate in the process of recruiting the eligible recreational group for the following reasons: (1) absence rate; (2) having professional training in other sport disciplines; (3) a physician’s certificate of past fracture or surgery. 

The elite and recreational groups were recruited from a relatively homogeneous population. Two groups were different in their training intensity, training patterns, rigid workout schedules and competition experiences. The exclusion criteria in both groups were a history of previous surgery in the lower limbs, traumas or fractures of the lower limbs in the previous six months, leg length discrepancies and other musculoskeletal disorders, such as neuropathies, rheumatoid arthritis and calcaneal spurs. During the study, the age, height, body weight, and body mass index (BMI) of each participant were recorded. Basic anthropometric characteristics for both groups are presented in Table 1. According to Table 1, the characteristics of the two groups’ height, mass and BMI are significantly different after the inspection by a two-group student-*t* test with a confidence level of 95%. All experiments in this study followed the guidelines of the research ethics committee of National Taiwan University and the recommendations of the Declaration of Helsinki.

### 2.2. Instruments

The JC Mat optical plantar pressure analysis device (View Grand International Co., Ltd, New Taipei City, Taiwan) integrated with FPDS-Pro software was applied to collect the arch index (AI) and plantar pressure distributions (PPDs) of both feet [26]. The repeatability and reproducibility of the device have been confirmed in previous studies, and it has been applied in exploring the AI and PPDs of athletes in running and basketball disciplines in static and dynamic states [27,28]. The measuring technology and principle of the JC Mat are parallel to the operation principle of the Harris footprint-measuring instrument. The following are the main features of the JC Mat: (1) foot characteristics are easily and effectively recognized; (2) the PPD and footprint images match the weight calibration experimental results; (3) There are 25 sensors per square centimeter for measuring plantar pressure, therefore, there are 13,600 sensors on each side (32 × 17 cm) of the JC Mat; (4) the sensitive pressure sensing with a wide working area can display and mark the delicate plantar pressure image with round dots; (5) the pressure distribution of footprints and barefoot images can be captured immediately; and (6) the built-in FPDS-Pro software can be used to analyze the following parameters: the AI and PPD values, the center of gravity balance, toe angle, and footprint images.

### 2.3. PPDs Assessment

The time for each experiment was set from 4 PM to 6 PM on the same day. In order to ensure consistency and trustworthiness of the present research, each experiment was conducted three days before and after the regular training courses and competitions. All participants’ anthropometric measurements were conducted to obtain their BMI values during the experiments.

It was necessary to secure data on static footprints via brief trials of static upright standing, where participants were required to follow the steps below:Roll the two trouser legs above the knees to avoid clothing restricting the movement of the limbs.Stand barefoot on the JC Mat sensor mat with specific markings and measuring range.Stand with feet shoulder-width apart and distribute body weight evenly on feet to control and balance the center of gravity.Stand in a natural posture, with arms hanging vertically at sides.Face the experiment instructor. Look the instructor straight in the eye. Keep the body still and balanced and relax the whole body until the foot pressure measured by JC Mat does not change significantly.

When the participant reaches the condition in step 5, the JC Mat directly records the pressure distribution from the static footprint. In the dynamic measurement experiment, each participant was instructed to walk barefoot at their self-comfort speed according to their accustomed pace and gait [29,30,31] on a 4 m-long JC Mat built-in walkway to its end, to make a turn, and to return by nature gait. Multiple walking trials were completed until at least three take-offs for the left and right foot were correctly acquired (i.e., the sensing cushion with marks of the specific measuring range of the JC Mat was struck with a single foot). The experiment was terminated by pressing the stop button. Autosaving of the collected data was completed when the close button was pressed.

### 2.4. PPDs Data Analysis

The values of AI and PPD were analyzed on three anatomical regions and six subregions of both feet. The computer program (FPDS-Pro software, View Grand International Co, Ltd., New Taipei City, Taiwan) was used for managing digital images of the static and walking footprints. The software generated the first line (a vertical line) on the footprint image. The vertical line extends from the tip of the second toe to the center of the heel; then, it automatically generates tangent lines to the foremost and rearmost end of the footprint excluding the toes. JC Mat built-in software automatically formed four parallel lines perpendicular to the vertical line and divided the footprint into three equal regions (region A, B, and C) and six subregions (subregions 1, 2, 3, 4, 5, and 6). Regions A, B, and C of a footprint were defined as the forefoot, midfoot, and rearfoot regions, respectively. The six subregions divided from the three regions were defined in the order as (1) the lateral metatarsal bone (LM), (2) the lateral longitudinal arch (LLA), (3) the lateral heel (LH), (4) the medial metatarsal bone (MM), (5) the medial longitudinal arch (MLA) and (6) the medial heel (MH). The AI ratio method proposed by Cavanagh and Rodgers assumes that AI is calculated as the ratio of the area of the middle third of the footprint divided by the area of the entire footprint excluding the toes, i.e., AI = B/(A + B + C). According to the definition of Cavanagh and Rodgers, an AI lower than 0.21 is a high arch, an AI between 0.21 and 0.26 is a normal arch, and an AI higher than 0.26 is a flat arch [32].

### 2.5. Rearfoot Postural Alignment Assessment

After the PPDs assessment, each participant’s posterior view of the rearfoot postural alignment was examined. All participants were guided to stand over a 30 cm height platform and keep their feet with natural apart (about 12–15 cm). The posterior view of each participant’s rearfoot postural alignment image was obtained (with a minimum size of 754 pixels and 96-ppi screen resolution) with a digital camera. According to the literature by Ribeiro et al. [33], the method of calculating the rearfoot static angle is as follows: to confirm that the rearfoot of both feet relaxed stand on the same horizontal line, and to determine the anatomical points in the lower back area of the legs: (1) the posterior calcaneal tuberosity; (2) the second point above the center of the calcaneus; and (3) the lower third of the leg. In the three-point connection, two lines were automatically generated by the Biomech 2019 postural analysis software (Loran Engineering SrL, Emilia-Romagna, Italy). The first standard straight of the lower extremity (a solid line) was drawn, which originated from the lower third of the leg to the calcaneal center. The second flip angle line of the lower extremity (a dotted line) was drawn, which originated from the posterior calcaneal tuberosity to the center of the calcaneus (Figure 1). The static rearfoot alignment was measured from the frontal alignment of a digital image by the software. The intersection of the extensions of both straight lines resulted in angles, which were classified as a normal foot (0° to 5°), varus (<0°), or valgus (>5°) [34].

### 2.6. Pain Assessment and Self-Reported Health Status

The elite rugby athletes’ soft-tissue pain and skeleton arrangement assessments and the self-reported health-status examination were implemented by a physiotherapist in the Rehabilitation Department of Taipei General Hospital after the plantar pressure measurement. This was essential for meeting the participant recruitment criteria, assessing physical symptoms, and identifying pain areas. The statements of the inquiry into the self-reported health status comprised the participants’ medical history, physical examination and general personal data.

In the process of the soft-tissue pain and skeleton arrangement assessments, the lower-limb pain was defined as the musculoskeletal pain which occurred during the past month and originated from the structures of the foot, ankle, knee, lower leg, and thigh. The definition excluded intermittent cramps, dermatological conditions, digital calluses, and nighttime paresthesia from the analysis. A standardized protocol of the questioning and examination techniques was used for ensuring the precise nature of the complaint. Physical examinations were conducted to evaluate the elite rugby athletes’ frequent soft-tissue and bone pains in their lower limbs based on the following steps:The physiotherapist examined the elite rugby athletes’ self-reported health status and pain complaints, and guided them to stand with bare feet and roll their trouser legs up to above the knees.The physiotherapist examined the elite rugby athletes’ lower extremities by palpating and pressing their feet (including navicular bones, cuboid bones, phalanges, metatarsals, and calcaneus), ankles, patella, knees, hips, tibias, fibulas and femur according to the participants’ self-reported health status, and re-examined the corresponding position on the other side of the pain areas. The physiotherapist then assessed the skeletal arrangement of the athletes’ lower limbs.In order to confirm the athletes’ pain areas precisely, the physiotherapist examined the following specific parts of the athletes’ common pain areas: (1) soft tissues, e.g., the plantar fascia, the Achilles tendon, the gastrocnemius, the tibialis anterior and posterior, the biceps, the quadriceps femoris, the medial and lateral ankle ligaments, the anterior cruciate ligaments, the medial and lateral collateral ligaments, the abductor hallucis and abductor digiti minimi of plantar plate and the lower back; (2) bone tissues of both feet, i.e., navicular bones, cuboid bones, phalanges, metatarsals, and calcaneus; (3) the ankles; (4) the patella; (5) the knees; (6) the hips; and (7) tibias.

### 2.7. Statistical Analysis

Descriptive statistics were used for outlining all participants’ ages, heights, weights and BMI values. Numerical data within this study are presented as mean and standard deviation (e.g., mean ± SD). The parameters of AI values, the PPDs of the forefoot, the midfoot, and the hindfoot three regions, and the PPDs of the six distinct subregions were compared between the groups using the independent sample *t*-test. In the study, the statistical significance was defined as *p* < 0.05 (marked as *) and *p* < 0.01 (marked as **). Statistical software (SPSS version18; SPSS Inc., Chicago, Illinois, USA) was used to manage the statistical analysis.

## 3. Results

### 3.1. Arch Index

Compared with the recreational group, the average bipedal AI value of the elite group was found to be significantly higher. Findings showed that the elite group’s arch type fell into the category of low arch (Table 2).

### 3.2. PPDs of the Three Regions in Static and Dynamic States

The PPDs were presented as percentages of the relative loads. In static standing, the relative load on the forefoot and midfoot regions of both feet in the elite group were significantly higher than that of the recreational group. No significant difference was observed in the rearfoot region (Table 3). 

During the midstance phase of walking, the elite group had a significantly higher relative load in the forefoot region, but lower in the rearfoot region compared with the recreational group (Table 4).

### 3.3. PPDs of the Six Subregions in Static Standing

The relative loads of the six subregions were secured from the data of three equal regions. Findings showed that the elite group’s relative loads in static standing were mainly concentrated on the entire forefoot and lateral midfoot regions. Compared with the recreational group, the elite group’s relative loads of the six subregions were significantly higher at the lateral metatarsals (left foot: 29.42% ± 2.74%; right foot: 29.43% ± 3.73%; *p* < 0.01) and the medial metatarsals (left foot: 22.41% ± 2.44%; right foot: 23.02% ± 3.53%; *p* < 0.05) of both feet. In the midfoot region, the elite group had a significantly higher relative load at the lateral longitudinal arch of both feet (left foot: 25.33% ± 4.22%; right foot: 24.59% ± 4.54%; *p* < 0.05). In the rearfoot region, the relative load at the medial heel of the elite group (left foot: 7.78% ± 1.96%; right foot: 8.74% ± 3.63%; *p* < 0.05) was significantly lower than that of the recreational group (Figure 2). 

### 3.4. PPDs of the Six Subregions during the Midstance Phase of Walking

Findings from the midstance phase of walking indicated that the elite group’s relative loads were mainly transferred to the forefoot region. Compared with the recreational group, the elite group’s relative loads of the six subregions were exerted more on the medial metatarsal bone of both feet (left foot: 30.89% ± 3.17%; right foot: 31.97% ± 2.70%; *p* < 0.01) and the lateral metatarsal bone of the right foot (30.29% ± 4.09%; *p* < 0.05). Nonetheless, in terms of the medial heel of the left foot, the relative loads of the elite group (7.44% ± 0.93%; *p* < 0.01) were significantly lower than those of the recreational group (Figure 3).

### 3.5. Footprint Image Characteristics

Compared with the recreational group, the static footprints of the elite group showed higher pressure distribution in the forefoot and midfoot regions of the feet (Figure 4). 

### 3.6. Rearfoot Postural Alignment Assessment

With regard to the changes in the bipedal rearfoot angle, the findings showed that the values of the static rearfoot alignment in the elite group conformed to the varus posture pattern (Table 5).

### 3.7. Pain Assessment and Self-Reported Health Status of the Participants

Based on findings from the pain assessment, the top 14 most common bone pain areas and soft tissue pains of the elite group were listed in descending order of percentage proportion in Table 6.

## 4. Discussion

The present research aimed not only at exploring the characteristics of plantar pressure with foot posture of elite rugby athletes and recreational rugby players during static standing and walking, but also at assessing the changes in pressure profiles which may develop lower-limb pains related to rugby drills. Previous studies maintained that the AI from footprints could be used for predicting the foot arch height and classifying footprint morphology [32,35,36,37]. Notably, the AI value used for defining the foot type category within this study was based on the literature by Cavanagh and Rodgers which observed that the AI value of normal arches was in the 0.21–0.26 range [32]. Nevertheless, findings from the present study were slightly different from those presented by Cavanagh and Rodgers. In their research, Cavanagh and Rodgers highlighted the critical value of AI (i.e., a mean AI of 0.23) calculated in the context of 107 young adults (mean age 30 years) without any foot symptoms. In the present study, the critical value of static AI of both feet in the recreational group was 0.21, which was calculated from an average of 57 male college and university students in Taiwan (mean age 20 years) with BMI (24.3 ± 0.5) and without musculoskeletal disorders of extremities. The difference in the normal range may inevitably exist in the subjective judgments of researchers, assessing equipment, characteristics of subjects and sample size; nonetheless, the study was set within the identical research condition. Based on the findings, the AI values of the feet in static standing were considerably symmetrical to each other within the respective groups. Recreational rugby players in the present research appeared to have normal foot arches. The elite rugby athletes’ AI value was higher than that of the recreational rugby players; therefore, the elite rugby athletes may have low-arched feet.

After analyzing the plantar loadings from barefoot static standing and the midstance phase of walking, three regional PPDs of the elite rugby athletes’ feet in static standing were found to be particularly focused on the forefoot and midfoot regions. However, during the midstance phase of walking, the plantar loads were mainly transferred to the forefoot region of both feet. This may be due to the fact that elite rugby league athletes tended to have posteriorly unbalanced posture under static conditions to conform to the balance of the foot and coordination of the whole body. However, in terms of the dynamic state, the results of this study appeared to support the studies by Gobbi et al. who found that rugby athletes kept forward-moving and that their heel loads were steadily reduced [15]. In their studies, Wong et al. observed that when performing four soccer-related movements, the athletes received higher pressure in their hallux, medial and the center of the forefoot [38].

In addition, a detailed evaluation of six subregional PPDs extended from the three regional PPDs were summarized as follows: in static standing, the plantar load of both feet of the elite rugby league athletes were mainly exerted on the medial and lateral metatarsals, and the lateral longitudinal arch. The plantar load on the medial heel was found to be relatively low. However, the plantar loads of elite rugby league athletes were dominantly transferred to the medial metatarsals of both feet and the lateral metatarsals of the right foot during the midstance phase of walking. In the aspect of the rearfoot postural alignment measurement, elite rugby league athletes’ bipedal rearfoot angle in static state were conformed to the rearfoot varus posture. The pattern of the results seemed to constitute a supination foot feature. The findings seem to be consistent with previous research by Ripani et al., who noted that rugby athletes’ right foot appeared to be supinated in static conditions, and the plantar pressure on the lateral area of the right foot was obviously higher [4]. The higher plantar load on the lateral foot may be related to difficulties in controlling the stability of the foot, or may be due to an increase in body weight resulting in an increase in triceps surae muscle tension to conform to the special requirements of the discipline. Differences in plantar load on the lateral metatarsals of both feet in the dynamic state may be attributed to the training tasks for their dominant leg. From the literature on similar exercises, Queen et al. argued that compared with other athletic tasks, soccer players have higher pressure peaks in the hallux, medial and middle forefoot regions, and that the highest pressure occurred in the middle forefoot region when performing acceleration tasks [39]. Findings from this study echoed the research by Sims et al. who noted that the plantar loads, maximum force and force–time integral in male soccer players were mainly beneath the middle portion of the forefoot as well as there being an increase in the contact area in lateral forefoot and midfoot regions during the cross-over cut task [40]. In studies by Sims et al., the medial portion of the foot (medial forefoot and midfoot) of female soccer players experience increased loading during acceleration and lateral cutting tasks. The increase in plantar loading on the lateral portion of the forefoot and midfoot regions in male soccer players could be associated with the increased incidence of fifth metatarsal injuries [40]. Research evidence by Lee et al. showed that under weight-bearing conditions, athletes with a history of stress fractures on the fifth metatarsal tended to have inverted rearfoot alignment [41]. An increased inversion of the rearfoot in basketball players is a common accident during cutting maneuvers [42]. A further explanation for the foot varus tendency of rugby players is that due to the special requirements of the athletes—muscle tension and tropism: as we all know, the action of the triceps surae muscle determines the rearfoot varus with the stiffness of the tarsal joints, and supination of the entire foot [43]. Excessive supination of the foot is considered to be due to an increase in calcaneal varus, which usually helps to reduce contact time during running [44]. Hasegawa et al. noted that runners suffered from a great degree of heel inversion when their feet were allowed to have a short contact time on the ground, and that a shorter contact time and a higher frequency of inversion during foot contact usually resulted in a higher running economy [44]. Therefore, the deformation of the foot arches seems to be inevitable for mechanical energy stored and released, force transmission, shock absorption, and, in particular, high impact sports, e.g., jumping and sprinting [45]. Similar results were observed in the studies by Ripani et al., who stressed that the dynamic conditions during the sports’ specific drills made rugby athletes exert considerably high pressure on the ground [4]. Rugby has a particular performance model, which, in turn, may make the lower extremities bear heavy loads which are placed on the feet. Owing to their high body weight together with the need to fix their feet on the ground and the press involving jumping and moving, rugby athletes generally have high pressures on their feet and a large foot–ground impact under dynamic conditions during the sports’ drills [4]. These researchers’ statements may explain the fact that findings from the elite rugby league athletes’ BMI in this study exceeded the normal healthy weight range of 18.5 to 22.9 defined by the World Health Organization (WHO) and the Asia-Pacific Guidelines.

Rugby players are reported to be at high risk of trauma and pathology associated with musculoskeletal training [46,47]. In this study, the most common bone pains of the elite rugby league athletes mainly occurred in the foot, including the plantar phalanx 1st, the metatarsophalangeal joint 1st and 2nd and 4th and 5th, the plantar metatarsal bone 1st and 2nd and 4th and 5th, and the calcaneus, etc. In addition, soft tissue pain most often occurs in the plantar plate (abductor hallucis and abductor digiti minimi), the medial and lateral collateral ligaments, the anterior cruciate ligaments, etc. The results seemed to support studies on exercises of similar nature, such as highly repetitive cutting maneuvers in basketball competition which may cause soft tissue injuries, such as ACL injuries [48], ankle sprains [49] and foot problems [50]. Extensive stop-and-go movements and cutting maneuvers in basketball games seem to be at the core of putting the knee ligaments and meniscus in danger [51]. Related studies have observed that male college basketball and soccer players have a high rate of ACL injury, which is the most common ligament rupture in the knee joint. Noncontact ACL injuries were common in basketball competition with rapid deceleration and rotational movements [52]. In some cases, basketball players usually land on another athlete’s foot, which can cause plantar flexion and varus, and can stretch the lateral ankle ligaments beyond their capacity. This, in turn, may result in an ankle sprain [53]. Exploring the elite rugby athletes’ plantar load distributions, Stovitz and Coetzee found that the forefoot abducts through medial rays and increases strength, which may cause problems with the first metatarsal bone (such as hallux valgus) and the second metatarsal bone (such as metatarsalgia) [54]. The pressure load applied to the forefoot area may cause Achilles tendinitis [23]. In addition, lateral ankle sprain to a certain degree is a distinct incidence involving the rearfoot supination and the lower leg’s external rotation. This mechanism is often described as a plantar flexion–inversion which frequently involves internal rotation (adduction) of the foot [55]. 

From the result of static footprints of both feet in elite rugby players, their plantar loadings were mainly applied to the forefoot region and the lateral longitudinal arch, but lower on the medial heel. Ankle inversion sprain may be attributed to the footprint pattern which can result in cuboid syndrome. More specifically, the pathological mechanism of the cuboid syndrome may originate from the cuboid valgus from the inverted foot position, such as the injury mechanism of the lateral ankle sprain, and the pain on the lateral column of the foot [56]. As evidenced in many studies, cuboid syndrome to a certain degree is related to other injuries that have signs and symptoms of the lateral aspect of the foot.

The limitation of this study may lie in the particular focus on plantar loading patterns of 51 elite and 57 recreational rugby players who were 19- to 22-year-old college or university athletes in Taiwan. Results from this research may inevitably limit the possibilities for generalization. It is widely acknowledged that strong athletes who specialize in collision sports generally have high body weight, and that body weight is usually considered to be one of the main factors related to changes in the shape of the arch [57]. All participants’ BMI within this study exceeded the normal range. Notably, however, the study was set in a relatively homogeneous population and under identical research condition. Considering the research setting and design, it can be argued that the participants’ BMI did not take a significant role in affecting the results from this study. Moreover, the issues of whether the participants had preferred/nonpreferred legs were not included within this research, due to the research design. However, it is worth further study to understand whether rugby athletes have a dominant leg based on a considerably larger number of research samples, and to explore how dominant legs may affect plantar pressure distribution, in particular. Given that findings of the walking plantar loads in the dynamic experiments were gained in the circumstances under which the participants were walking at their self-comfort speed on the device, differences in the participants’ walking speed may inevitably exist in the process of the experiments. This, in turn, could affect the data collected by the researchers to a certain extent. Furthermore, it is important to consider the use of an algometer for further study to quantify the threshold of musculoskeletal pain assessment by replacing general physical examination. Notably, however, few studies currently explore the plantar load characteristics of recreational and elite rugby athletes by centering on the difference between static standing and dynamic states. Findings from this research could shed light on the static and dynamic plantar pressure distribution and lower-limb pain profiles of the Taiwanese college and university elite rugby athletes. It is expected that the results may reinforce the possible link between plantar pressure distribution and pain characteristics. Arguably, the results concerning the elite rugby athletes’ plantar pressure characteristics can be constructive and useful for the related specialists in medical practices and the field of plantar healthcare in the process of developing rugby boots and orthopedic insoles for buffering uneven plantar loading, improving footwear comfort and reducing sports-related injuries of rugby athletes.

## 5. Conclusions

Elite rugby league athletes in this study were generally classified as having low arches. The elite rugby league athletes’ characteristics of plantar pressure distributions and foot posture revealed that the higher plantar loads mainly distributed on the forefoot and midfoot, accompanied by rearfoot varus in bipedal static stance. The higher plantar load mainly exerted on the forefoot during the midstance phase of walking. The lower-limb pain profiles echoed the literature on rugby injuries, and this could serve as the traceable beginning for the possible links among low-arched supinated feet, metatarsophalangeal joint pains and cuboid syndrome of elite rugby league athletes.

## Figures and Tables

**Figure 1 ijerph-19-01158-f001:**
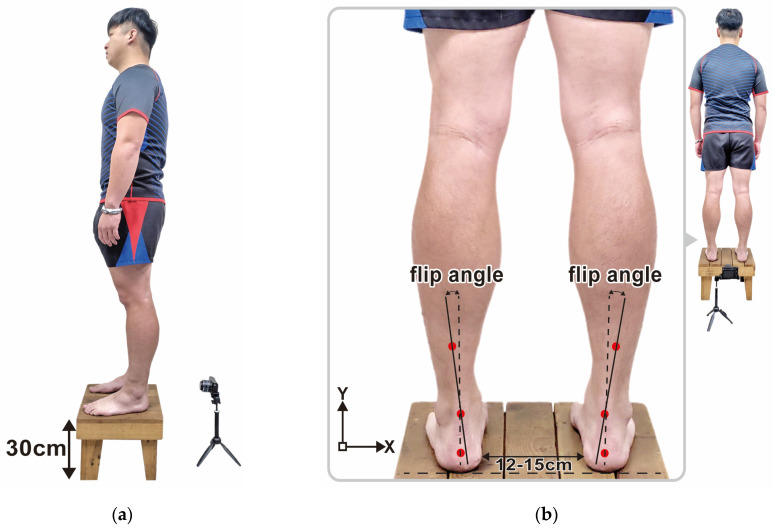
Schematic diagram of each participant in a static standing posture in (**a**) the lateral and (**b**) posterior view, coupled with positioning marks in the back of lower legs and measurement methods of the rearfoot alignment.

**Figure 2 ijerph-19-01158-f002:**
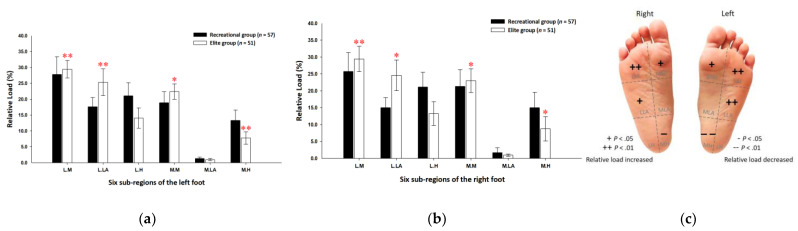
The plantar pressure distributions of the left (**a**) and right (**b**) feet in the six subregions in static standing. Description of changes in pressure distribution is illustrated in (**c**) plantar diagram. Through the independent sample *t*-test, * *p* < 0.05 and ** *p* < 0.01 were significantly different between both groups. The six subregions and their abbreviations are as follows: LH, lateral heel; LLA, lateral longitudinal arch; LM lateral metatarsal bone; MH, medial heel; MLA, medial longitudinal arch; and MM, medial metatarsal bone.

**Figure 3 ijerph-19-01158-f003:**
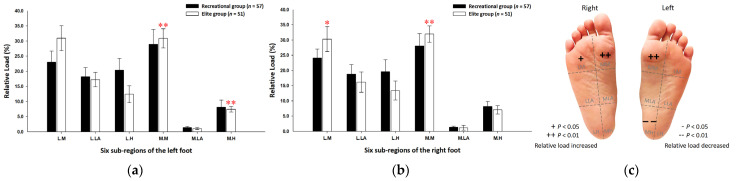
The plantar pressure distributions of the left (**a**) and right (**b**) feet in the six subregions during the midstance phase of walking. Description of changes in pressure distribution is illustrated in (**c**) plantar diagram. Through the independent sample *t*-test, * *p* < 0.05 and ** *p* < 0.01 were significantly different between both groups. The six subregions and their abbreviations are as follows: LH, lateral heel; LLA, lateral longitudinal arch; LM lateral metatarsal bone; MH, medial heel; MLA, medial longitudinal arch; and MM, medial metatarsal bone.

**Figure 4 ijerph-19-01158-f004:**
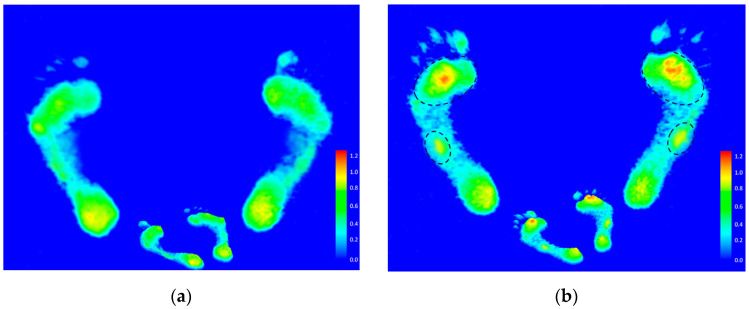
The static footprint image of the recreational group (**a**) and the elite group (**b**). Black circle indicates the areas of higher pressure.

**Table 1 ijerph-19-01158-t001:** Anthropometric characteristics of the groups.

Characteristic	Recreational Group ^1^ (n = 57)	Elite Group ^2^ (n = 51)
Age (years)	19.6 ± 1.1	20.4 ± 1.3
Height (cm)	172.3 ± 4.4	179.0 ± 4.7 *
Mass (kg)	72.1 ± 3.7	86.0 ± 5.8 *
BMI (m/kg)	24.3 ± 0.8	26.9 ± 1.6 *
Rugby training experience (years)	4.3 ± 1.0	5.3 ± 0.9

Abbreviation: BMI, body mass index (calculated as the weight in kilograms divided by the square of the height in meters). Note: Values are given as mean ± SD. * *p* < 0.05 (student-*t* test, 2-tails). ^1^ Healthy eligible recreational rugby players (the recreational group) were age-matched male college and university students. ^2^ Elite rugby league athletes (the elite group) were male college and university students who were qualified first-division rugby athletes and had more than five successive years of rugby training experience in the National Rugby Championship, Taiwan Rugby Championship 7 s and Taiwan Chung-Cheng Cup-Senior High School Group.

**Table 2 ijerph-19-01158-t002:** Arch Index of the Foot in Static Standing.

	Recreational Group	Elite Group	*p* Value ^1^
Left foot	0.21 ± 0.04	0.30 ± 0.05	0.039
Right foot	0.21 ± 0.04	0.31 ± 0.05	0.033

Note: Data are given as mean ± SD. ^1^
*p* values were determined by the independent sample *t*-test between the recreational group (n = 57) and the elite group (n = 51).

**Table 3 ijerph-19-01158-t003:** Relative Load of the Forefoot, Midfoot, and Rearfoot Regions in Static Standing.

Region	Recreational Group	Elite Group	*p* Value ^1^
Left foot			
Forefoot (%)	23.34 ± 6.40	25.91 ± 4.37	<0.01
Midfoot (%)	9.49 ± 8.43	13.16 ± 12.59	<0.01
Rearfoot (%)	17.17 ± 5.39	10.93 ± 4.10	0.058
Right foot			
Forefoot (%)	23.50 ± 5.74	26.23 ± 4.84	<0.05
Midfoot (%)	8.34 ± 7.09	12.77 ± 12.31	<0.01
Rearfoot (%)	18.07 ± 5.41	11.00 ± 4.22	0.074

Note: Data are given as mean ± SD. ^1^
*p* values were determined by the independent sample *t*-test between the recreational group (n = 57) and the elite group (n = 51).

**Table 4 ijerph-19-01158-t004:** Relative Load of the Forefoot, Midfoot, and Rearfoot Regions During the Midstance Phase of Walking.

Region	Recreational Group	Elite Group	*p* Value ^1^
Left foot			
Forefoot (%)	25.98 ± 5.24	30.91 ± 3.65	<0.01
Midfoot (%)	9.82 ± 8.72	9.16 ± 8.28	0.154
Rearfoot (%)	14.22 ± 6.95	9.94 ± 3.25	<0.01
Right foot			
Forefoot (%)	26.04 ± 4.05	31.13 ± 3.55	<0.01
Midfoot (%)	10.06 ± 9.03	8.64 ± 7.92	0.200
Rearfoot (%)	13.91 ± 6.47	10.23 ± 3.99	<0.01

Note: Data are given as mean ± SD. ^1^
*p* values were determined by the independent sample *t*-test between the recreational group (n = 57) and the elite group (n = 51).

**Table 5 ijerph-19-01158-t005:** Static Rearfoot Postural Alignment.

	Recreational Group	Elite Group	*p* Value ^1^
Left foot	2.48 ± 4.57	−1.30 ± 5.45	0.042
Right foot	2.25 ± 3.52	−2.45 ± 5.78	<0.01

Note: Data are given as mean ± SD. ^1^
*p* values were determined by the independent sample *t*-test between the recreational group (n = 57) and the elite group (n = 51).

**Table 6 ijerph-19-01158-t006:** Pain Assessment and Self-Reported Health Status in the 51 Elite Rugby Athletes.

Pain Area	Elite Group (No. [%])	Pain Area	Elite Group (No. [%])
Bone pain		Soft-tissue pain	
Foot (Plantar phalanx 1st)	47 (92.2)	Plantar plate (Abductor hallucis)	47 (92.2)
Foot (Metatarsophalangeal joint 1st & 2nd)	46 (90.2)	Plantar plate (Abductor digiti minimi)	47 (92.2)
Foot (Plantar metatarsal bone 1st & 2nd)	46 (90.2)	Medial collateral ligament (MCL)	36 (70.6)
Foot (Metatarsophalangeal joint 4th & 5th)	43 (84.3)	Lateral collateral ligament (LCL)	32 (62.7)
Foot (Plantar metatarsal bone 4th & 5th)	43 (84.3)	Anterior cruciate ligament (ACL)	31 (60.8)
Foot (Calcaneus)	37 (72.5)	Lateral ankle ligament	27 (52.9)
Medial knee joint	36 (70.6)	Medial ankle ligament	25 (49.0)
Lateral knee joint	32 (62.7)	Quadriceps femoris	23 (45.1)
Lateral ankle joint	32 (62.7)	Biceps femoris	21 (41.2)
Patella	31 (60.8)	Tibialis anterior	19 (37.3)
Medial ankle joint	25 (49.0)	Gastrocnemius	19 (37.3)
Tibia	19 (37.3)	Achilles tendon	18 (35.3)
Hip joint	13 (25.5)	Plantar fascia	15 (29.4)
Others	6 (11.8)	Lower back	15 (29.4)

## Data Availability

The datasets generated and/or analyzed during the current study are available from the corresponding author upon reasonable request.

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
