# Peer review of "Characteristics of Plantar Pressure with Foot Postures and Lower Limb Pain Profiles in Taiwanese College Elite Rugby League Athletes"

_ijerph, 2022, doi:10.3390/ijerph19031158_

Round 1

Reviewer 1 Report

Dear coauthors, 

I apreciated the changes made to the manuscript, as suggested in the las review.

In my opinion, the article is now suitable for publication.

Regards.

Author Response

Dear Reviewer,

Coauthors and I appreciated the reviewers’ constructive suggestions and constructive comments on our manuscript (ID: ijerph-1518451). The suggestions and comments are helpful for improving our manuscript. We are submitting the revised version of the manuscript with our responses to the suggestions and comments by the reviewers.

Reviewer 2 Report

I appreciate the improvements made by the authors. But I think, there are still aspects to highlight some minor comments;

  • The author should to add which test was used to analyze the homogenity of data in statistical analysis section.
  • Also, the authors must to explain how they find the sample size for each group?

Author Response

Dear Reviewer,

Coauthors and I appreciated the reviewers’ constructive suggestions and constructive comments on our manuscript (ID: ijerph-1518451). The suggestions and comments are helpful for improving our manuscript. We are submitting the revised version of the manuscript with our responses to the suggestions and comments by the reviewers.

Our responses to each suggestion and comment are as follows, and they are also presented in red text with a grey background color in the revised manuscript:

Reviewer 3 Report

This study is a significant study that clarified the relationship between foot pressure distribution and pain in rugby players.
However, the author has already reported a similar study (https://www.mdpi.com/1660-4601/18/24/12942), and I had to consider whether it constituted a double submission.
As a result, I have determined that this study is different in content from the previously reported paper and that the following comments are sufficient to warrant acceptance.

Discussion
In line 441, the author mentions the relationship with ACL and MCL injuries, but this study did not analyze the kinematics of the knee joint, and there seems to be a leap in discussion. In order to consider the relationship between ACL injury and MCL injury, it is desirable to analyze the movement of subjects with abnormal foot pressure distribution and point out the relationship.

Author Response

Dear Reviewer,

Coauthors and I appreciated the reviewers’ constructive suggestions and constructive comments on our manuscript (ID: ijerph-1518451). The suggestions and comments are helpful for improving our manuscript. We are submitting the revised version of the manuscript with our responses to the suggestions and comments by the reviewers.

This manuscript is a resubmission of an earlier submission. The following is a list of the peer review reports and author responses from that submission.

Round 1

Reviewer 1 Report

This study is mostly about the foot problems of rugby players. Even though the article is well written and well designed, it might be out of scope. In addition, I have some comments.  

  • Abstract: The abstract is starting with “This study aimed at exploring the correlation between…” but the study was not included any correlation analysis. Also, it is necessary to present the study design. In addition, it is interesting to situate the reader in relation to the characteristics of the population, for example, age, BMI, facilitating the understanding of the application of the study in clinical practice.
  • The introduction is good and leaved the reader to the main point of the study but it can be better to give more information or comparison about plantar pressure and arch index base on literature. The authors previously published two articles (ref 28,29) in different sports with the same methodology, and they did not mention or discussed them.
  • The authors should explain more about the musculoskeletal pain assessment. The experience of PT is enough for palpation but how was standardized the force pressure of palpations? It might be good to use an algometer for these assessments?
  • How many participants were initially evaluated to select the 51/57? What was the drop-out rate? Were the participants taken any medication (AINS, analgesics, etc?)
  • If the authors had dominant leg (preferred/non-preferred leg) information of athletes, please mention in the method. Dominancy can affect plantar pressure disturbition, please add this in charecteristics (Table 1) and method section ( and explain how is identified).
  • Is there any reliability study about the JC Mat system for measuring the outcomes?
  • The authors must provide a reference to the affirmation about JC Mat specifications (line 135-145)
  • The walking procedures in measurements of footprint need some references (Line 166-169). The walking speed is seems to be a important factor changing foot pressure, If not, please add this information as a study limitation.
  • The author should to explain which test was used to analyze the homogenity of data in statistical analysis. Also, the authors did not present a sample size calculation. Did you perform power analysis to find the sample size for each group?
  • The results are clear and well presented, There is no need to represent of characteristics of groups in Table 1 abbreviations.
  • The discussion can be expanded. The authors concluded the elite player demonstrated low medial heel pressure on both test conditions however there were no reflections or differences in lateral heel pressure? Or what is the reason for LLA pressure changes in walking than standing in elite rugby player? Also what is the reason for the plantar pressure distributions differences of the left and right feet in LM and MH during walking. Additionally, Did the BMI difference between athletes (groups) influenced your results? Because there can be some adaptation in muscle and joint mechanics in human body, please read this article “Usgu, S., Ramazanoğlu, E., & Yakut, Y. (2021). The Relation of Body Mass Index to Muscular Viscoelastic Properties in Normal and Overweight Individuals. Medicina, 57(10), 1022”

Reviewer 2 Report

First of all, to thank the effort and dedication in contributing to the updating of the scientific field through the generation of publications.

In my opinion, it has a very original and interesting choice of subject matter, however it does not reflect an adequate scientific methodology in its development for the following reasons:

- There is no properly specified research methodology. For example: type of study (observational, descriptive, longitudinal analytical, etc ...)

Participant Recruitment:

The sex of the participants is not clear from the study. If they are all men (or women), they should specify why they have not considered sex as a study variable. Anthropometric characteristics linked to sex can be binding.The authors are kindly requested to clarify this in the study.

Instruments and Equipment:

A plantar pressure platform is used at work. It is not specified which one, nor if it is scientifically validated. The reproducibility and repeatability conditions are very important. This premise is necessary for the viability of the results.

Bibliography:

More than 80% of the bibliographic citations used are more than 8 years old. Of these, more than 60% are older than 10 years and some are also more than 20 years old.It is considered necessary to update the bibliography for a good scientific documentation.

Reviewer 3 Report

I think it is a very interesting work and deals with a subject that may interest readers. But before accepting your publication I recommend that authors take into account these considerations:

  • The abstrac must be reformulated, it does not reflect all the sections of the work, it must also be a detailed description, as it is very abstract.
  • I consider that the descriptions in the introduction of motor adaptation and complex reaction time should not be included. These concepts should be known to expert readers.
  • Why do the authors refer (30) to the object of their work?... this should be reviewed
  • Section 2.1. it would be more advisable to expose it in a more graphic way.
  • Section 2.2. it must be reformulated, it is not necessary that they describe in detail the evidence that has previously been published and that the authors cite them. It would be enough to name and describe the objective and reference the quote where it is described.
  • The results should be clearly stated directly and chronologically, focusing on the main findings.
  • The discussion should contain further comments on the authors' assessments, as well as expose how these results could be put into practice.
  • The conclusions must clearly answer the objective set based on the results obtained in the work, so I do not understand why the authors make references in this section.